# TURNING THE CURSE OF HETEROGENEITY IN FEDERATED LEARNING INTO A BLESSING FOR OUT-OF-DISTRIBUTION DETECTION

**Shuyang Yu[1], Junyuan Hong[1], Haotao Wang[2], Zhangyang Wang[2] and Jiayu Zhou[1]**
[1]Department of Computer Science and Engineering, Michigan State University
[2]Department of Electrical and Computer Engineering, University of Texas at Austin
`{yushuyan,hongju12,jiayuz}@msu.edu, {htwang,atlaswang}@utexas.edu`

## ABSTRACT

Deep neural networks have witnessed huge successes in many challenging prediction tasks and yet they often suffer from out-of-distribution (OoD) samples, misclassifying them with high confidence. Recent advances show promising OoD detection performance for centralized training, and however, OoD detection in federated learning (FL) is largely overlooked, even though many security sensitive applications such as autonomous driving and voice recognition authorization are commonly trained using FL for data privacy concerns. The main challenge that prevents previous state-of-the-art OoD detection methods from being incorporated to FL is that they require large amount of real OoD samples. However, in real-world scenarios, such large-scale OoD training data can be costly or even infeasible to obtain, especially for resource-limited local devices. On the other hand, a notorious challenge in FL is data heterogeneity where each client collects non-identically and independently distributed (non-iid) data. We propose to take advantage of such heterogeneity and turn the curse into a blessing that facilitates OoD detection in FL. The key is that for each client, non-iid data from other clients (unseen external classes) can serve as an alternative to real OoD samples. Specifically, we propose a novel Federated Out-of-Distribution Synthesizer (FOSTER), which learns a class-conditional generator to synthesize virtual external-class OoD samples, and maintains data confidentiality and communication efficiency required by FL. Experimental results show that our method outperforms the state-of-the-art for OoD tasks by 2.49%, 2.88%, 1.42% AUROC, and 0.01%, 0.89%, 1.74% ID accuracy, on CIFAR-10, CIFAR-100, and STL10, respectively. Codes are available: https://github.com/illidanlab/FOSTER.

## 1 INTRODUCTION

Deep neural networks (DNNs) have demonstrated exciting predictive performance in many challenging machine learning tasks and have transformed various industries through their powerful prediction capability. However, it is well-known that DNNs tend to make overconfident predictions about what they do not know. Given an out-of-distribution (OoD) test sample that does not belong to any training classes, DNNs may predict it as one of the training classes with high confidence, which is doomed to be wrong (Hendrycks & Gimpel, 2016; Hendrycks et al., 2018; Hein et al., 2019).

To alleviate the overconfidence issue, various approaches are proposed to learn OoD awareness which facilitates the test-time detection of such OoD samples during training. Recent approaches are mostly achieved by regularizing the learning process via OoD samples. Depending on the sources of such samples, the approaches can be classified into two categories: 1) the *real-data* approaches rely on a large volume of real outliers for model regularization (Hendrycks et al., 2018; Mohseni et al., 2020; Zhang et al., 2021); 2) the *synthetic* approaches use ID data to synthesize OoD samples, in which a representative approach is the virtual outlier synthesis (VOS) (Du et al., 2022).

While both approaches are shown effective in centralized training, they cannot be easily incorporated into federated learning, where multiple local clients cooperatively train a high-quality centralized

model without sharing their raw data (Konečnỳ et al., 2016), as shown by our experimental results in Section 5.2. On the one hand, the real-data approaches require substantial real outliers, which can be costly or even infeasible to obtain, given the limited resources of local clients. On the other hand, the limited amount of data available in local devices is usually far from being sufficient for synthetic approaches to generate effective virtual OoD samples.

Practical federated learning approaches often suffer from the curse of heterogeneous data in clients, where non-iid (Li et al., 2020b) collaborators cause a huge pain in both the learning process and model performance in FL (Li et al., 2020a). Our key intuition is to turn the curse of data heterogeneity into a blessing for OoD detection: The heterogeneous training data distribution in FL may provide a unique opportunity for the clients to communicate knowledge outside their training distributions and learn OoD awareness. A major obstacle to achieving this goal, however, is the stringent privacy requirement of FL. FL clients cannot directly share their data with collaborators. This motivates the key research question: *How to learn OoD awareness from non-iid federated collaborators while maintaining the data confidentiality requirements in federated learning?*

In this paper, we tackle this challenge and propose Federated Out-of-distribution SynThesizER (FOSTER) to facilitate OoD learning in FL. The proposed approach leverages non-iid data from clients to synthesize virtual OoD samples in a privacy-preserving manner. Specifically, we consider the common learning setting of class non-iid (Li et al., 2020b), and each client extracts the external class knowledge from other non-iid clients. The server first learns a virtual OoD sample synthesizer utilizing the global classifier, which is then broadcast to local clients to generate their own virtual OoD samples. The proposed FOSTER promotes diversity of the generated OoD samples by incorporating Gaussian noise, and ensures their hardness by sampling from the low-likelihood region of the class-conditional distribution estimated. Extensive empirical results show that by extracting only external-class knowledge, FOSTER outperforms the state-of-out for OoD benchmark detection tasks.

The main contributions of our work can be summarized as follows:

- We propose a novel federated OoD synthesizer to take advantage of data heterogeneity to facilitate OoD detection in FL, allowing a client to learn external class knowledge from other non-iid federated collaborators in a privacy-aware manner. Our work bridges a critical research gap since OoD detection for FL is currently not yet well-studied in literature. To our knowledge, the proposed FOSTER is the first OoD learning method for FL that does not require real OoD samples.

- The proposed FOSTER achieves the state-of-art performance using only limited ID data stored in each local device, as compared to existing approaches that demand a large volume of OoD samples.

- The design of FOSTER considers both the diversity and hardness of virtual OoD samples, making them closely resemble real OoD samples from other non-iid collaborators.

- As a general OoD detection framework for FL, the proposed FOSTER remains effective in more challenging FL settings, where the entire parameter sharing process is prohibited due to privacy or communication concerns. This is because that FOSTER only used the classifier head for extracting external data knowledge.

## 2 RELATED WORK

**OoD detection.** Existing OoD detection methods are mainly from two complementary perspectives. The first perspective focused on post hoc. Specifically, Hendrycks & Gimpel (2016) first introduced a baseline utilizing maximum softmax distribution probabilities (MSP). Based in this work, many improvements have been made by follow-up works in recent years, such as the calibrated softmax score (ODIN) (Liang et al., 2017), Mahalanobis distance (Lee et al., 2018), energy score (Liu et al., 2020), Likelihood Regret (Xiao et al., 2020), Confusion Log Probability (CLP) score (Winkens et al., 2020), adjusted energy score Lin et al. (2021), k-th nearest neighbor (KNN) (Sun et al., 2022), and Virtual-logit Matching (ViM) (Wang et al., 2022). Compared with post hoc methods, FOSTER can dynamically shape the uncertainty surface between ID and OoD samples. Different post hoc methods are also applied in our experiment section as baselines.

Another perspective tends to detect OoD samples by regularization during training, in which OoD samples are essential. The OoD samples used for regularization can be either real OoD samples or

virtual synthetic OoD samples. Real OoD samples are usually natural auxiliary datasets (Hendrycks et al., 2018; Mohseni et al., 2020; Zhang et al., 2021). However, real OoD samples are usually costly to collect or infeasible to obtain, especially for terminals with limited sources. Regularization method utilizing virtual synthetic OoD samples do not rely on real outliers. Grcić et al. (2020) trained a generative model to obtain the synthetic OoD samples. Jung et al. (2021) detect samples with different distributions by standardizing the max logits without utilizing any external datasets. Tack et al. (2020); Sehwag et al. (2021) proposed contrastive learning methods that also does not rely on real OoD samples. Du et al. (2022) proposed VOS to synthesize virtual OoD samples based on the low-likelihood region of the class-conditional Gaussian distribution. Current state-of-the-art virtual OoD methods are usually thirsty for ID data, which is not sufficient enough for local clients. Compared with these existing methods, the proposed FOSTER can detect OoD samples with limited ID data stored in each local device, without relying on any auxiliary OoD datasets.

**Federated Learning.** Federated learning (FL) is an effective machine learning setting that enables multiple local clients to cooperatively train a high-quality centralized mode (Konečný et al., 2016). FedAvg (McMahan et al., 2017), as a classical FL model, performs model averaging of distributed local models for each client. It shows an excellent effect on reducing the communication cost. Based on FedAvg, many variants (Wang & Joshi, 2018; Basu et al., 2019) have been proposed to solve the problems arising in FedAvg, such as convergence analysis (Kairouz et al., 2021; Qu et al., 2020), heterogeneity (Li et al., 2020a; Hsu et al., 2019; Karimireddy et al., 2020; Zhu et al., 2021), communication efficiency (Reddi et al., 2020). Among these problems, although heterogeneity of data will make the performance of ID data worse, it will give us a great chance to learn from the external data from other non-iid collaborators. Even though FOSTER is used the FedAvg framework, as a general OoD detection method for FL, FOSTER can also be applied to other variants of FedAvg.

## 3 PROBLEM FORMULATION

In this paper, we consider classification tasks in heterogeneous FL settings, where non-iid clients have their own label set for training and testing samples. Our goal is to achieve OOD-awareness on each client in this setting.

**OoD training**. The OoD detection problem roots in general supervised learning, where we learn a classifier mapping from the instance space $\mathcal{X}$ to the label space $\mathcal{Y}$. Formally, we define a learning task by the composition of a data distribution $\mathcal{D} \subset \mathcal{X}$ and a ground-truth *labeling* oracle $c^* : \mathcal{X} \to \mathcal{Y}$. Then any $x \sim \mathcal{D}$ is denoted as in-distribution (ID) data, and otherwise, $x \sim \mathcal{Q} \subset \mathcal{X} \backslash \mathcal{D}$ as out-of-distribution data. Hence, an ideal OoD detection oracle can be formulated as a binary classifier $q^*(x) = \mathbb{I}(x \sim \mathcal{D})$, where $\mathbb{I}$ is an indication function yielding 1 for ID samples and $-1$ for OoD samples. With these notations, we define the OoD learning task as $\mathcal{T} := \langle \mathcal{D}, \mathcal{Q}, c^* \rangle$.

To parameterize the labeling and OoD oracles, we use a neural network consisting of two stacked components: a feature extractor $f : \mathcal{X} \to \mathcal{Z}$ governed by $\boldsymbol{\theta}^f$, and a classifier $h : \mathcal{Z} \to \mathcal{Y}$ governed by $\boldsymbol{\theta}^h$, where $\mathcal{Z}$ is the latent feature space. For the ease of notation, let $h_i(z)$ denote the predicted logit for class $i = 1, \ldots, c$ on extracted feature $z \sim \mathcal{Z}$. We unify the parameters of the classifier as $\boldsymbol{\theta} = (\boldsymbol{\theta}^f, \boldsymbol{\theta}^h)$. We then formulate the OoD training as minimizing the following loss on the task $\mathcal{T}$:

$$J_{\mathcal{T}}(\boldsymbol{\theta}) := \mathbb{E}_{x \sim \mathcal{D}} \left[ \ell_{\text{CE}} \left( h(f(x; \boldsymbol{\theta}^f); \boldsymbol{\theta}^h), c^*(x) \right) \right] + \lambda \, \mathbb{E}_{x' \sim \mathcal{Q}} \left[ \ell_{\text{OE}} \left( f(x'; \boldsymbol{\theta}^f); \boldsymbol{\theta}^h \right) \right],$$

where $\ell_{\text{CE}}$ is the cross-entropy loss for supervised learning and $\ell_{\text{OE}}$ is for OoD regularization. We use $\mathbb{E}[\cdot]$ to denote the expectation estimated by the empirical average on samples in practice. The non-negative hyper-parameter $\lambda$ trade off the OoD sensitivity in training. We follow the classic OoD training method, Outlier Exposure (Hendrycks et al., 2018), to define the OoD regularization for classification problem as

$$\ell_{\text{OE}}(z'; \boldsymbol{\theta}^h) := E(z'; \boldsymbol{\theta}^h) - \sum_{i=1}^{c} h_i(z'; \boldsymbol{\theta}^h), \tag{1}$$

where $E(z'; \boldsymbol{\theta}^h) = -T \log \sum_i^c e^{h_i(z'; \boldsymbol{\theta}^h)/T}$ is the energy function, given the temperature parameter $T > 0$. At test time, we approximate the OoD oracle $q^*$ by the MSP score (Hendrycks & Gimpel, 2016).

**Heterogeneous federated learning** (FL) is a distributed learning framework involving multiple clients with non-iid data. There are different non-iid settings (Li et al., 2020b; 2021), and in this paper,

we follow a popular setting that the non-iid property is only concerned with the classes (Li et al., 2020b). Given $K$ clients, we define the corresponding set of tasks $\{\mathcal{T}_k\}_{k=1}^K$ where $\mathcal{T}_k = \langle \mathcal{D}_k, \mathcal{Q}_k, c_k^* \rangle$ and $c_k^* : \mathcal{X} \to \mathcal{Y}^k$ are non-identical for different $k$ resulting non-identical $\mathcal{D}_k$. Each $\mathcal{Y}^k$ is a subset of the global label set $\mathcal{Y}$. Since the heterogeneity is known to harm the convergence and performance of FL (Yu et al., 2020), we adopt a simple personalized FL solution to mitigate the negative impact, where each client uses a personalized classifier head $h_k$ upon a global feature extractor $f$ (Arivazhagan et al., 2019). This gives the general objective of FL: $\min_{\boldsymbol{\theta}} \frac{1}{K} \sum_{k=1}^K J_{\mathcal{T}_k}(\boldsymbol{\theta})$. The optimization problem can be solved alternatively by two steps: 1) local minimization of the objective on local data and 2) aggregation by averaging client models. In this paper, we assume that each client only learns classes they see locally during training, because updating classifier parameters for unseen classes has no data support and doing so will almost certainly harm the performance of FL. To see this, Diao *et al.* showed that masking out unseen classes in the cross-entropy loss can merit the FL training (Diao et al., 2020).

**Challenges**. When we formulate the OoD training in FL, the major challenge is defining the OoD dataset $\mathcal{Q}_k$, which does not come for free. The centralized OoD detection of VOS assumes $\mathcal{Q}_k$ is at the tail of an estimated Gaussian distribution of $\mathcal{D}_k$ (Du et al., 2022), which requires enormous examples from $\mathcal{D}_k$ for an accurate estimation of parameters. However, such a requirement is usually not feasible for a client *per se*, and the construction of $\mathcal{Q}_k$ remains a challenging question.

## 4 METHOD

In this section, we first introduce the intuition of our proposed FOSTER, then elaborate on how to synthesize virtual external class data and avoid the hardness fading of the virtual OoD samples. The proposed framework is illustrated in Fig. 1.

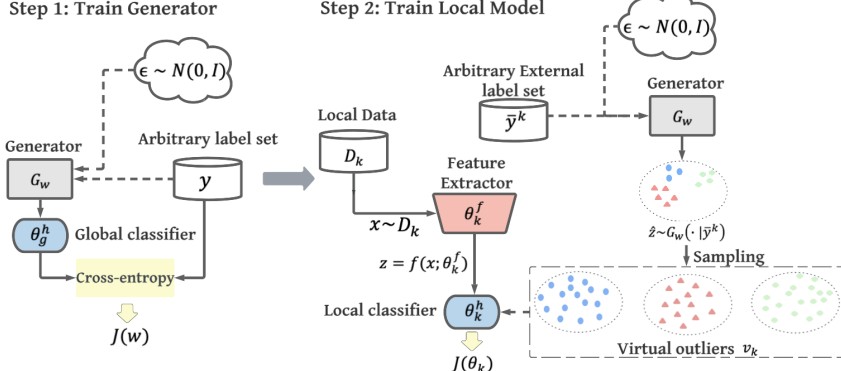

Figure 1: The framework of FOSTER. In step 1, to extract external class knowledge from local clients, the server first trains a generator utilizing the global classifier based on a cross-entropy objective function $J(\mathbf{w})$ (Eq. (2)). In step 2, each local client utilizes the generator received to generate their own external class data $z$. To preserve the hardness of the virtual OoD samples, we also sample virtual outliers $\mathbf{v}_k$ from the low-likelihood region of the class-conditional distribution estimated for the generated OoD samples. The virtual OoD samples $\mathbf{v}_k$ are used for regularization of local client objective $J(\boldsymbol{\theta}_k)$ (Eq. (5)).

### 4.1 NATURAL OoD DATA IN NON-IID FL

Recent advances show promising OoD detection performance by incorporating OoD samples during the training phase, and however, OoD detection in FL is largely overlooked. In FL, each client does not have access to a large volume of real OoD samples because it can be costly or even infeasible to obtain such data for resource-constrained devices. As such, an OoD training method for FL that relies on few or even no real OoD examples is strongly desired. Novel to this work, we notice that data from classes out of the local class set, namely *external-class data*, are natural OoD samples *w.r.t.* the local data and can serve as OoD surrogate samples in OoD training. As shown in Fig. 2, training w/ *external-class data* achieves better OoD detection performance than normal training and

VOS, since the score of ID and real OoD data is well separated. Besides, compared to the real OoD dataset adopted in prior arts, external-class samples are likely to be nearer to the ID data, since they are sampled from similar feature distributions (refer to (a) and (b) in Fig. 2 ).

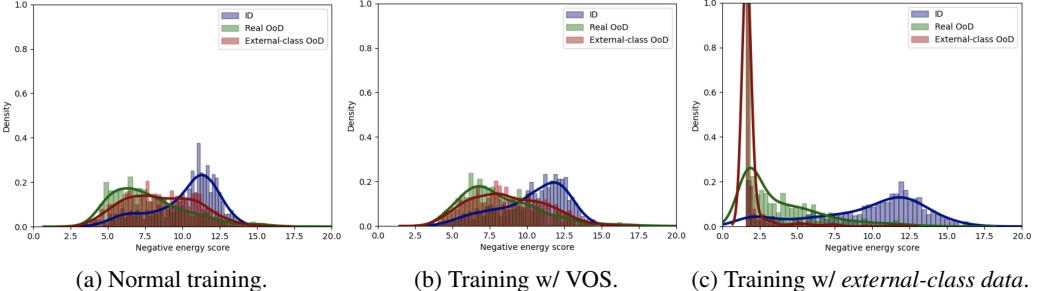

(a) Normal training.     (b) Training w/ VOS.     (c) Training w/ *external-class data*.

Figure 2: The density of negative energy score for OoD detection evaluation using dataset Textures. We use 5 ID classes, and 5 external classes of CIFAR-10.

### 4.2 Synthesizing External-Class Data from Global Classifier

Though using external-class data as an OoD surrogate is attractive and intuitive, it is not feasible in FL to directly collect them from other non-iid clients, due to privacy concerns and high communication costs on data sharing.

We thereby propose to generate samples from the desired classes leveraging the encoded class information in the global classifier head. Given the global classifier $H : \mathcal{Z} \to \mathcal{Y}$ parameterized by $\boldsymbol{\theta}_g^h$, we utilize a $\mathbf{w}$-governed conditional generative network $G_{\mathbf{w}} : \mathcal{Y} \to \mathcal{Z}$ to generate samples from specified classes on clients' demand. As such, we solve the following optimization problem:

$$\min_{\mathbf{w}} J(\mathbf{w}) := \mathbb{E}_{y \sim p(y)} \mathbb{E}_{z \sim G_{\mathbf{w}}(z|y,\epsilon)} \left[ \ell_{\mathrm{OE}}(H(z; \boldsymbol{\theta}_g^h), y) \right], \tag{2}$$

where $p(y)$ is the ground-truth prior which is assumed to be a uniform distribution here. We follow the common practice of the generative networks (Zhu et al., 2021) to let $y$ be a one-hot encoding vector, where the target class entry is 1 and others are 0. To encourage the diversity of the generator outputs $G(z|y)$, we use a Gaussian noise vector $\epsilon \sim \mathcal{N}(0, I)$ to reparameterize the one-hot encoding vector during the generating process, following the prior practice (Kingma & Welling, 2013). Thus, $G_{\mathbf{w}}(z|y) \equiv G_{\mathbf{w}}(y, \epsilon|\epsilon \sim \mathcal{N}(0, I))$ given $y \sim \mathcal{Y}$, where $\mathcal{Y}$ is the global label set. The generator training process can refer to Fig. 1 Step 1. Then for local training (see Fig. 1 Step 2), by downloading the global generator as a substitute of $\mathcal{Q}_k$, each local client indexed by $k$ can generate virtual OoD samples given an arbitrary external class set $\bar{\mathcal{Y}}^k = \mathcal{Y} \backslash \mathcal{Y}^k$. In the feature space, we denote the virtual samples as $z \sim G_{\mathbf{w}}(z|y, \epsilon)$ given $y \sim \bar{\mathcal{Y}}^k$.

### 4.3 Filtering Virtual External-Class Samples

Although synthesized features are intuitively conditioned on external class, the quality of generated OoD samples may vary by iterations likely because of the lack of two properties: (1) *Diversity*. Like traditional generative models (Srivastava et al., 2017; Thanh-Tung & Tran, 2020), the trained conditional generator may suffer from mode collapse (Mao et al., 2019) in a class, where generator can only produce a small subsets of distribution. As a result, the effective synthesized OoD samples will be mostly unchanged and OoD training will suffer from the lack of diverse samples. (2) *Hardness*. For a client, its internal and external classes may co-exist with another client, which will enlarge the between-class margins gradually. As the FL training proceeds, the class-conditioned synthesis OoD samples will become increasingly easier to be memorized, namely, overfit by the model. In other words, the hardness of OoD examples declines over time.

**(1) Encourage OoD diversity by tail sampling**. As mode collapse happens in the high-density area of a class, samples that approximate the class but have larger variance are preferred for higher diversity. For this purpose, we seek to find samples of low but non-zero probability from the distribution of the external classes. Specifically, for each client, we first assume that the set of virtual OoD representations $\{z_{ki} \sim G(z|y_i, \epsilon)|y_i \sim \bar{\mathcal{Y}}^k, \epsilon \sim \mathcal{N}(0, I)\}_{i=1}^{N_k}$ forms a class-conditional multivariate Gaussian distribution $p(z_k|y_k = c) = \mathcal{N}(\boldsymbol{\mu}_k^c, \boldsymbol{\Sigma}_k^c)$, where $\boldsymbol{\mu}_k^c$ is the Gaussian mean of samples from

the external class set $\bar{\mathcal{Y}}^k$ for client $k$, and $\boldsymbol{\Sigma}_k^c$ is the tied covariance matrix. The parameters of the class-conditional Gaussian can be estimated using the empirical mean and variance of the virtual external class samples:

$$\hat{\boldsymbol{\mu}}_k^c = \frac{1}{N_k^c} \sum_{i:y_i=c} z_{ki}, \quad \hat{\boldsymbol{\Sigma}}_k = \frac{1}{N_k} \sum_c \sum_{i:y_i=c} (z_{ki} - \hat{\boldsymbol{\mu}}_k^c)(z_{ki} - \hat{\boldsymbol{\mu}}_k^c)^T, \tag{3}$$

where $N_k$ is the number of samples, and $N_k^c$ is the number of samples of class $c$ in the virtual OoD set. Then, we select the virtual outliers falling into the $\epsilon$-likelihood region as:

$$\mathcal{V}_k^c = \{\mathbf{v}_k^c | \varepsilon_0 < \frac{\exp\left(-\frac{1}{2}(\mathbf{v}_k^c - \hat{\boldsymbol{\mu}}_k^c)^T \hat{\boldsymbol{\Sigma}}_k^{-1}(\mathbf{v}_k^c - \hat{\boldsymbol{\mu}}_k^c)\right)}{(2\pi)^{d/2}|\hat{\boldsymbol{\Sigma}}_k|^{1/2}} < \varepsilon, \mathbf{v}_k^c \sim G(\cdot|y = c, \epsilon)\}, \tag{4}$$

where $\varepsilon_0$ ensures the sample is not totally random, a small $\varepsilon$ pushes the generated $\mathbf{v}_k^c$ away from the mean of the external class in favor of the sampling diversity.

---

**Algorithm 1** Federated Out-of-Distribution Synthesizer (FOSTER)

---

1: **Input:** Tasks $\{\mathcal{T}_k\}_{k=1}^K$;
   Global parameters $\boldsymbol{\theta}_g$, local parameters $\{\boldsymbol{\theta}_k\}_{k=1}^K$;
   Global generator parameter $\mathbf{w}$;
   Learning rate $\alpha$, $\beta$, local steps $T$, ID batch size $B$, OE batch size $B_{OE}$.
2: **repeat**
3:     Server selects active clients $\mathcal{A}$ uniformly at random, then broadcast $\boldsymbol{\theta}$, $\mathbf{w}$ to $\mathcal{A}$.
4:     **for** all user $k \in \mathcal{A}$ in parallel **do**
5:         Initialize local parameters $\boldsymbol{\theta}_k \leftarrow \boldsymbol{\theta}$
6:         **for** $t = 1, \ldots, T$ **do**
7:             $\{(x_i, y_i)\}_{i=1}^B \sim \mathcal{D}_k, Z_{OE} = \{z_{ki} \sim G(z|y_i, \epsilon)|y_i \sim \bar{\mathcal{Y}}^k, \epsilon \sim \mathcal{N}(0, I)\}_{i=1}^{B_{OE}}$.
8:             Estimate the multivariate Gaussian distributions based on $Z_{OE}$ by Eq. (3).
9:             Filter virtual external class samples according to Eq. (4).
10:        $\boldsymbol{\theta}_k \leftarrow \boldsymbol{\theta}_k - \beta \nabla_{\boldsymbol{\theta}_k} J(\boldsymbol{\theta}_k)$.   $\triangleright$ Optimize Eq. (5)
11:       **end for**
12:       Client $k$ sends $\boldsymbol{\theta}_k$ back to the server.
13:     **end for**
14:     Server updates $\boldsymbol{\theta}_g \leftarrow \frac{1}{|\mathcal{A}|} \sum_{k \in \mathcal{A}} \boldsymbol{\theta}_k$.
15:     **for** $t = 1, \ldots, T$ **do**
16:       $\mathbf{w} \leftarrow \mathbf{w} - \alpha \nabla_{\mathbf{w}} J(\mathbf{w})$.   $\triangleright$ Optimize Eq. (2)
17:     **end for**
18: **until** training stop

---

**(2) Increase the hardness by soft labels**. To defend the enlarged margin between internal and external classes, we control the condition inputs to the generator such that generated samples are closer to the internal classes. Given an one-hot encoding label vector $y$ of class $c$, we assign $1 - \delta$ to the $c$-th entry, and a random value within $(0, \delta)$ to the rest of the positions, where $\delta \in (0, 0.5)$.

In summary, given an observable $\hat{\mathcal{D}}_k$, we formulate the local optimization of FOSTER as:

$$\min_{\boldsymbol{\theta}_k} J(\boldsymbol{\theta}_k) := \frac{1}{|\hat{\mathcal{D}}_k|} \sum_{x_i \in \hat{\mathcal{D}}_k} \left[ \ell_{\text{CE}}(h_k(f(x_i; \boldsymbol{\theta}_k^f); \boldsymbol{\theta}_k^h), c^*(x_i)) + \lambda \frac{1}{|\mathcal{V}_k|} \sum_{\mathbf{v}_k \in \mathcal{V}_k} \ell_{\text{OE}}(\mathbf{v}_k) \right], \tag{5}$$

and the overall framework of our algorithm is summarized in Algorithm 1. The major difference from FedAvg is that we introduce a generator for OoD outlier synthesis. Since the generator is trained on the server, the computation overhead for the client is marginal, with only the inference of low-dimensional vectors. As compared to VOS, the samples generated from external classes are more likely to approximate the features from real images due to the supervision of real external class prototypes from the classifier head.

## 5 EXPERIMENTS

In this section, we first introduce the experiment setup and then show empirical results demonstrating the effectiveness of the proposed FOSTER.

**ID Datasets for training**. We use CIFAR-10, CIFAR-100 (Krizhevsky et al., 2009), STL10 (Coates et al., 2011), and DomainNet (Peng et al., 2019) as ID datasets. Both CIFAR-10 and CIFAR-100 are large datasets containing 50, 000 training images and 10, 000 test images. Compared with CIFAR, STL10 is a small dataset consisting of only 5,000 training images and 8,000 test images. DomainNet are consist of images from 6 different domains. We use Domain-Net to explore how FOSTER performs in the case of feature non-iid among different clients.

**OoD Datasets for evaluation**. We use Textures (Cimpoi et al., 2014), Places365 (Zhou et al., 2017), LSUN-C (Yu et al., 2015), LSUN-Resize (Yu et al., 2015) and iSUN (Xu et al., 2015) as the OoD datasets for evaluation. When ID dataset is CIFAR-10, we also evaluate on CIFAR-100 to check near-OoD detection performance, since CIFAR-10 and CIFAR-100 datasets have similarities, although their classes are disjoint.

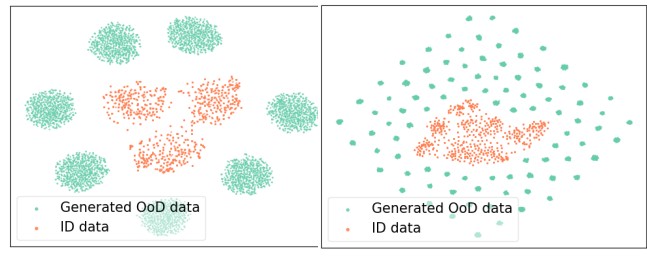

(a) CIFAR-10.    (b) CIFAR-100.

Figure 3: Visualization of generated external class samples and ID samples.

**Baselines**. We compare the proposed FOSTER with both Post hoc and Virtual synthetic OoD detection methods that have been mentioned in Section 2: a) Post hoc OoD detection methods: Energy score (Liu et al., 2020), MSP (Hendrycks & Gimpel, 2016), ODIN (Liang et al., 2017). b) Synthetic OoD detection method: VOS (Du et al., 2022). For a fair comparison, the training method of FL for all of the above approaches including the proposed FOSTER is FedAvg (McMahan et al., 2017) with a personalized classifier head, and we note that our framework can be extended to other FL variants. All the approaches only use ID data without any auxiliary OoD dataset for training.

**Metrics for OoD detection and classification**. To evaluate the classification performance on ID samples, we report the test accuracy (Acc) for each client's individual test sets, whose classes match their training sets. For OoD detection performance, we report the area under the receiver operating characteristic curve (AUROC), and the area under the PR curve (AUPR) for ID and OoD classification. In FL setting, all three metrics are the mean results of all the clients.

**Heterogeneous federated learning**. For CIFAR-10 and CIFAR-100, the total client number is 100, for STL10, the total client number is 50. For DomainNet, the total client number is 12 (2 for each domain). To model class non-iid data of the training datasets, we follow a uniform partition mode and assign partial classes to each client. We distribute 3 classes per client for CIFAR-10 and STL10, 5 classes for DomainNet, and 10 classes for CIFAR-100, unless otherwise mentioned.

### 5.1 Visualization of generated external class samples

In Fig. 3, we visualize the generated external class samples and ID samples of a client using TSNE for both CIFAR-10 and CIFAR-100. Without accessing the raw external-class data from the other users, our generator, trained merely from the shared classifier head, yields samples that are strictly out of the local distribution without any overlap. We also obtain a consistent conclusion from CIFAR-100, which has as many as 90 external classes. The enormous external classes diversify the OoD set and therefore we observe a larger gain of OoD detection accuracy (a 2.9% AUROC increase versus the best baseline) compared to other benchmarks in Table 1. The observation also motivates our design of the tail sampling to encourage diversity.

### 5.2 Benchmark Results

**FOSTER outperforms existing methods.** We compare FOSTER with other competitive baselines in Table 1. The proposed FOSTER shows stronger OoD detection performance on all three training sets, while preserving a high test accuracy. VOS is another regularization method using virtual OoD samples, which even shows worse results than post hoc methods. The virtual OoD data synthesized by VOS is based on a large amount of ID samples. For the FL setting, when data stored in each device is limited, these synthesized OoD samples based on ID data will no longer be effective, which

deteriorates the OoD detection performance. For FOSTER, the virtual OoD samples are based on the external class knowledge extracted from other clients, which are close to real OoD samples. Thus, they are effective in improving the OoD detection performance while preserving the test accuracy.

| ID dataset | Method | Acc ↑ | AUROC ↑ | AUPR ↑ |
|---|---|---|---|---|
| CIFAR-10 | Energy | 0.9431 | 0.7810 | 0.9262 |
| | MSP | 0.9431 | 0.8829 | 0.9691 |
| | ODIN | 0.9431 | 0.8842 | 0.9689 |
| | VOS | 0.9426 | 0.7970 | 0.9342 |
| | FOSTER | **0.9432** | **0.9091** | **0.9785** |
| CIFAR-100 | Energy | 0.8129 | 0.8056 | 0.9575 |
| | MSP | 0.8129 | 0.8606 | 0.9782 |
| | ODIN | 0.8129 | 0.8657 | 0.9789 |
| | VOS | 0.8063 | 0.8372 | 0.9666 |
| | FOSTER | **0.8218** | **0.8945** | **0.9838** |
| STL10 | Energy | 0.8236 | 0.7529 | 0.9228 |
| | MSP | 0.8236 | 0.7410 | 0.9309 |
| | ODIN | 0.8236 | 0.7418 | 0.9306 |
| | VOS | 0.8264 | 0.7370 | 0.9126 |
| | FOSTER | **0.8410** | **0.7671** | **0.9425** |

Table 1: Our FOSTER outperforms competitive baselines. ↑ indicates larger value is better. **Bold** numbers are best performers.

**Near OoD detection.** We evaluate the model training with CIFAR10 datasets on both near OoD (CIFAR100) and far OoD datasets. The results are shown in Table 2, and the best results are highlighted. The proposed FOSTER outperforms baselines for all of the evaluation OoD datasets, especially the near OoD dataset CIFAR100. By synthesizing virtual external class samples, FOSTER has access to virtual near OoD samples during training, which is also an advantage of FOSTER over other baselines.

| Datasets | Textures | | Places365 | | LSUN-C | | LSUN-Resize | | iSUN | | CIFAR-100 | |
|---|---|---|---|---|---|---|---|---|---|---|---|---|
| | AUROC | AUPR | AUROC | AUPR | AUROC | AUPR | AUROC | AUPR | AUROC | AUPR | AUROC | AUPR |
| Energy | 0.7080 | 0.8868 | 0.8221 | 0.9411 | 0.7009 | 0.9065 | 0.8376 | 0.9519 | 0.8289 | 0.9462 | 0.7883 | 0.9248 |
| MSP | 0.8107 | 0.9375 | 0.8964 | 0.9754 | 0.9043 | 0.9774 | 0.9154 | 0.9825 | 0.9103 | 0.9805 | 0.8604 | 0.9615 |
| ODIN | 0.8124 | 0.9367 | 0.8976 | 0.9752 | 0.9062 | 0.9773 | 0.9166 | 0.9825 | 0.9114 | 0.9805 | 0.8614 | 0.9613 |
| VOS | 0.7346 | 0.8993 | 0.8267 | 0.9447 | 0.7270 | 0.9196 | 0.8451 | 0.9541 | 0.8397 | 0.9499 | 0.8086 | 0.9379 |
| FOSTER | **0.8458** | **0.9544** | **0.9253** | **0.9842** | **0.9332** | **0.9863** | **0.9316** | **0.9870** | **0.9238** | **0.9849** | **0.8952** | **0.9742** |

Table 2: Near and far OoD detection for CIFAR10. The proposed FOSTER outperforms baselines for all of the evaluation OoD datasets, especially near OoD dataset CIFAR100.

**OoD detection for feature non-iid clients.** We explore whether our FOSTER can still work well when feature non-iid also exists among different clients on Domain-Net. Under this problem setting, different clients not only have different classes, but may also come from different domains. According to the results shown in Table 3, although the results are not that significant compared with feature iid settings, FOSTER still outperforms the baselines. For feature non-iid settings, the external class knowledge extracted from clients from

| Method | Acc ↑ | AUROC ↑ | AUPR ↑ |
|---|---|---|---|
| Energy | 0.7237 | 0.6745 | 0.8953 |
| MSP | 0.7237 | 0.6871 | 0.9048 |
| ODIN | 0.7237 | 0.6871 | 0.9047 |
| VOS | 0.7340 | 0.6796 | 0.8988 |
| FOSTER | **0.7348** | **0.6960** | **0.9075** |

Table 3: Our FOSTER outperforms competitive baselines under feature non-iid setting.

different domains is not so consistent compared with feature iid cases. However, our experimental results also show that in this case, there is still some invariant external class information across different domains that can be extracted by our FOSTER to help improve the OoD detection performance.

## 5.3 QUALITATIVE STUDIES

**Effects of active client number.** We investigate the effects of active client number on CIFAR-10. The number of clients is fixed to be 100, while the number of active clients is set to be 20, 50 and 100, respectively. According to the results in Table 4, FOSTER shows better OoD detection performance than baselines in all cases of active users. With the increase of active clients, the OoD performance of FOSTER remains stable, which means our proposed FOSTER is not sensitive to the number of active users.

| Active num | Method | Acc ↑ | AUROC ↑ | AUPR ↑ | Classes / client | Method | Acc ↑ | AUROC ↑ | AUPR ↑ |
|---|---|---|---|---|---|---|---|---|---|
| 20 | Energy | 0.9399 | 0.7760 | 0.9363 | 10 | Energy | 0.8129 | 0.8056 | 0.9575 |
|  | MSP | 0.9399 | 0.8560 | 0.9674 |  | MSP | 0.8129 | 0.8606 | 0.9782 |
|  | ODIN | 0.9399 | 0.8562 | 0.9674 |  | ODIN | 0.8129 | 0.8657 | 0.9789 |
|  | VOS | **0.9410** | 0.7545 | 0.9173 |  | VOS | 0.8063 | 0.8372 | 0.9666 |
|  | FOSTER | 0.9401 | **0.9011** | **0.9776** |  | FOSTER | **0.8218** | **0.8945** | **0.9838** |
| 50 | Energy | 0.9432 | 0.7592 | 0.9185 | 5 | Energy | 0.8976 | 0.7735 | 0.9157 |
|  | MSP | **0.9432** | 0.8869 | 0.9728 |  | MSP | 0.8976 | 0.8776 | 0.9704 |
|  | ODIN | 0.9432 | 0.8879 | 0.9727 |  | ODIN | 0.8976 | 0.8831 | 0.9714 |
|  | VOS | 0.9430 | 0.7946 | 0.9311 |  | VOS | 0.8974 | 0.7927 | 0.9289 |
|  | FOSTER | 0.9429 | **0.8947** | **0.9750** |  | FOSTER | **0.8981** | **0.9081** | **0.9778** |
| 100 | Energy | 0.9431 | 0.7810 | 0.9262 | 3 | Energy | 0.9383 | 0.7215 | 0.8684 |
|  | MSP | 0.9431 | 0.8829 | 0.9691 |  | MSP | 0.9383 | 0.8682 | 0.9586 |
|  | ODIN | 0.9431 | 0.8842 | 0.9689 |  | ODIN | 0.9383 | 0.8723 | 0.9592 |
|  | VOS | 0.9426 | 0.7970 | 0.9342 |  | VOS | 0.9393 | 0.7636 | 0.8990 |
|  | FOSTER | **0.9432** | **0.9091** | **0.9785** |  | FOSTER | **0.9397** | **0.8865** | **0.9697** |

Table 4: Ablation study on the number of active clients:FOSTER is not sensitive to the number of active users.

Table 5: Ablation study on ID class number: the advantage of the proposed FOSTER over other baselines is not affected by the number of ID class number.

**Effects of ID class number.** We investigate the effects of ID class number on CIFAR-100. We set the classes distributed per client (classes / client) to be 10, 5 and 3, respectively. According to the results in Table 5, the advantage of the proposed FOSTER over other competitive baselines is not affected by the number of ID classes. When the number of ID classes decreases, for FOSTER the maximum changes in AUROC and AUPR are 2.16% and 0.81%, respectively. VOS, as another virtual synthetic OoD detection method, with the decrease of ID classes, AUROC and AUPR drop by 7.36% and 6.76%, respectively, which is a much larger variation compared with our method. Thus, the ID class number has a large impact on VOS, while almost has no effect on FOSTER.

**Effects of the p.d.f. filter.** We report the effects of the p.d.f. filter as mentioned in Section 4.3 on CIFAR-10 in Table 6. The generator without a p.d.f. filter is outperformed by baselines. The phenomenon occurs because not all generated external class samples are of high quality, and some of them may even deteriorate OoD detection performance. Since we add Gaussian noise during the process, some randomly generated external class samples might overlap with ID samples. Thus, we build a class-condition Gaussian distribution for external classes, and adopt a p.d.f. filter to select diverse virtual OoD samples which do not overlap with the ID clusters. According to this table, filtering out low-quality OoD samples improves AUROC and AUPR by 4.44% and 1.18%, respectively.

| Method | Acc ↑ | AUROC ↑ | AUPR ↑ |
|---|---|---|---|
| Energy | 0.9431 | 0.7810 | 0.9262 |
| MSP | 0.9431 | 0.8829 | 0.9691 |
| ODIN | 0.9431 | 0.8842 | 0.9689 |
| VOS | 0.9426 | 0.7970 | 0.9342 |
| FOSTER w/o pdf filter | 0.9425 | 0.8647 | 0.9667 |
| FOSTER w/ pdf filter | **0.9432** | **0.9091** | **0.9785** |

Table 6: Ablation study on pdf filter: pdf filter plays an effective role in selecting diverse, high-quality virtual OoD samples.

| Method | Acc ↑ | AUROC ↑ | AUPR ↑ |
|---|---|---|---|
| FOSTER | **0.8410** | 0.7671 | 0.9425 |
| FOSTER w/ soft label | 0.8294 | **0.7872** | **0.9501** |

Table 7: Ablation study on random soft labels: soft label strategy increase the hardness of generated virtual OoD samples.

**Effects of the random soft label strategy.** We study the effects of the random soft label strategy on STL10, and set $\delta = 0.2$. As shown in Table 7, after replacing the one-hot label with the random soft label as the input for the generator, we improve AUROC and AUPR by 2.01% and 0.75% respectively, while preserving a similar ID classification test accuracy. That is because soft label contains knowledge from ID classes make the generated external class samples closer to ID samples.

# 6 CONCLUSION

In this paper, we study a largely overlooked problem: OoD detection in FL. To turn the curse of heterogeneity in FL into a blessing that facilitates OoD detection, we propose a novel OoD synthesizer without relying on any real external samples, allowing a client to learn external class knowledge from other non-iid federated collaborators in a privacy-preserving manner. Empirical results showed that the proposed approach achieves state-of-the-art performance in non-iid FL.

ACKNOWLEDGMENTS

This material is based in part upon work supported by the National Science Foundation under Grant IIS-2212174, IIS-1749940, Office of Naval Research N00014-20-1-2382, and National Institute on Aging (NIA) RF1AG072449. The work of Z. Wang is in part supported by the National Science Foundation under Grant IIS-2212176.

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
