# OpenReview forum: "Turning the Curse of Heterogeneity in Federated Learning into a Blessing for Out-of-Distribution Detection"
_ICLR.cc/2023/Conference — ICLR 2023 notable top 25%_

### Official Review · Reviewer_1Z2K · 2022-10-22

**Confidence:** 4
**Clarity, Quality, Novelty And Reproducibility:** 1) The paper is clearly written, but …
**Correctness:** 4
**Technical Novelty And Significance:** 3
**Empirical Novelty And Significance:** 3
**Recommendation:** 8

**Strength And Weaknesses:**

Strength.

1) This paper explores the OoD detection method for federated learning for the first time, which is indeed a problem to be solved.

2) Without the need for additional real OoD samples, FOSTER can generate valid virtual OoD samples on clients with limited class data. Experimental evaluations show that this strategy outperforms previous OoD detection methods in federated learning scenarios, which is worthy of recognition.

Weaknesses.
1) The technical novelty of this paper is not good enough. The techniques used in the paper, including virtual sample generation, tail sampling, and soft labels, have been used in previous OoD detection. The main contribution is to use these technologies in the federated learning scenario.

2) The specific training process of the sample synthesizer lacks clarity. After reading Section 4.2 I still don't understand how synthesizers learn class information and synthesize samples efficiently. Hope for a clearer explanation.
Minors.

3) Figure 2 is hard to read.

4) Typo. See the last sentence of the 'challenge' paragraph.


**Summary Of The Paper:**

This paper proposes a virtual OoD sample synthesizer suitable for federated learning, called FOSTER. It uses the class information collected by the client to train the generator on the server and broadcast. Under the premise of ensuring privacy, FOSTER uses the knowledge of other non-iid federated partners to generate virtual OoD samples locally on the client. Using virtual samples to join training will improve the OoD detection performance of the model. Experimental results show that FOSTER outperforms traditional OoD detection methods in federated learning.

**Summary Of The Review:**

This paper explores the OoD detection technology under federated learning, proposes a virtual OoD synthesizer suitable for this scenario, and achieves performance beyond the baseline. I recognize the significance of the work and the performance gains achieved by the paper. But the paper lacks some novelty in technology, and the description of the method needs to be clearer.

---

> ### Author Response · Authors · 2022-11-18
> **Thank you for your helpful comments and suggestions**
>
> We are glad that the reviewer found our idea interesting. We thank the reviewer for the constructive comments and suggestions, which we address below:
>
> **1. The technical novelty of this paper is not good enough. The techniques used in the paper, including virtual sample generation, tail sampling, and soft labels, have been used in previous OoD detection. The main contribution is to use these technologies in the federated learning scenario.**
>
> Thanks for your careful reading! We agree that the building components in our method share similarities with previous techniques as you pointed out. However, we would like to humbly argue that this does not prevent our paper from making its significant contribution to the community.
>
> Our paper provides the first solution of tackling OoD detection in federated learning (FL). For the detection of OoD samples nowadays has been essential to protect the integrity of the predictive model (especially the deep ones), and yet the data confidentiality requirement of FL and the limited sample size of FL clients have imposed critical challenges of OoD detection in FL. This challenge prevents directly plugging in previous OoD detection techniques, as we have illustrated in our paper and empirical studies. Tackling this procedure is non-trivial, and we proposed a novel framework that utilizes external class data from other clients to train OOD-awareness in heterogeneous FL. To our knowledge, the proposed FOSTER is the first OoD learning method for FL that does not require real OoD samples.
>
> The unique challenge in this new setting is to simultaneously train OOD-awareness and preserve data privacy, as the latter is a basic requirement in FL. To overcome this challenge, we design a privacy-preserving external-class sample generator (Section 4.2) and a filtering method to improve the diversity of the selected OOD training data (Section 4.3). As a result, our final framework innovatively leveraged the data heterogeneity in FL to train OOD-awareness, turning the curse of data heterogeneity into a blessing that facilitates OoD detection.
>
> **2. The specific training process of the sample synthesizer lacks clarity.**
>
> Thanks for pointing out the clarity issue and we have addressed it by providing more details in our paper. We illustrated the training process in Fig.1. Specifically, the server first trains a generator to generate representations minimizing class-conditioned losses. Then, each local client utilizes the received generator to generate their own external class data. To preserve the hardness of the virtual OoD samples, we also sample virtual outliers from the low-likelihood region of the class-conditional distribution estimated for the generated OoD samples. Finally, the virtual OoD samples are used for the regularization of local client objective functions.
>
>
> **3. Figure 2 is hard to read.**
>
> We have revised Figure 2 to address the confusion. To better explain the intuition that external-class data are natural OoD data that can be used for OoD training, we change the original Fig.2 into the density of negative energy score for OoD detection evaluation using dataset Textures. In the updated Fig. 2, we compare the energy score used for evaluation after **normal training**, **training w/ VOS**, and **training w/ external-class data**. Results show that **training w/ external-class can effectively increase the margin between real OoD and ID data**. Fig.2 also verifies our other intuition that external-class samples are likely to be nearer to the ID data. The revised part is marked in blue in the manuscript. We hope the revised version is much clearer for readers.

---

> > ### Comment · Reviewer_1Z2K · 2022-11-22
> > **Thanks for your response**
> >
> > The author's response addressed my concerns about technical novelty and clarity.
> > I will increase my score.

---

> > > ### Author Response · Authors · 2022-11-23
> > > **Thank you for reading our response**
> > >
> > > Thank you very much for carefully reading our response and increasing your score! We are glad our response has addressed your concerns.

---

### Official Review · Reviewer_3v8K · 2022-10-23

**Confidence:** 3
**Correctness:** 2
**Technical Novelty And Significance:** 2
**Empirical Novelty And Significance:** 2
**Recommendation:** 6

**Clarity, Quality, Novelty And Reproducibility:**

Clarity: The motivation can be improved (i.e., Section 4.1). The paper should clearly demonstrate why external-class data can help in distinguishing the ID and OoD data.

Quality: The paper is in a good shape but not solid enough as it lacks in-depth analyses of the effectiveness of the proposed approach.

Novelty: The idea of using GAN to help the training is not new [1].

[1] FedCG: Leverage Conditional GAN for Protecting Privacy and Maintaining Competitive Performance in Federated Learning.

Reproducibility: Some experimental details are missing, e.g., what is the test dataset in the experiments.


**Strength And Weaknesses:**

Strength: The studied problem is important and less exploited in the current FL literature.

Weaknesses:

1. The intuition of FOSTER is to utilize the external-class samples as OoD. However, in the testing, the external-class samples are not true OoD samples. Then, the models may misclassify these samples into OoD. In the experiments, do the authors consider the external-class samples when testing the ID classification accuracy? If yes, why the ID classification accuracy is high given the external-class samples?

2. FOSTER is based on personalized FL. What about the baselines? The paper only mentions that the training method of all approaches is FedAvg in Section 5. The baselines should also adopt personalized FL for fair comparison. Also, local training without FL can also be added as a baseline, e.g., the clients can simply conduct unsupervised learning locally such as one-class SVM.

3. Figure 2 is confusing. How is the training performed? Is it based on OoD training or is it based on vanilla supervised training? The green points seem to be mixed with blue and yellow points even in the left figure.

4. A generator is required in FOSTER, which introduces additional privacy concern, communication, and computation overhead. The paper should discuss these aspects.

5. Experiments to investigate the effect of client sampling and data heterogeneity can be added.

6. Typo: Line 3 of Algorithm 1: broadcast \theta and w to A; Section 6: “we studied”, “we propose”. Please keep consistency for the tense.

**Summary Of The Paper:**

The paper proposes a new algorithm named FOSTER for out-of-distribution (OoD) detection in the federated setting. Since there may be no OoD sample in the federated setting, the paper proposes to use samples from the other classes that do not exist in the current client as OoD samples for local training. Since the raw data are not allowed to transfer, FOSTER trains a conditional GAN in the server, which helps to generate the synthetic external-class samples during local training. The experiments show that FOSTER outperforms the other approaches that directly apply existing OoD studies in the federated setting.

**Summary Of The Review:**

I think the paper still needs more analysis and experiments to justify the effectiveness of the proposed approach.

---

> ### Author Response · Authors · 2022-11-18
> **Thank you for your helpful comment and suggestions - Part 1**
>
> Thank you for your comment and we address your concerns as follows:
>
> **1. Do the authors consider the external-class samples when testing the ID classification accuracy? If yes, why the ID classification accuracy is high given the external-class samples?**
>
> Thank you for the question and we note that our current descriptions may have caused the confusion. We have added details to clarify this. Specifically, 1) for each client, its training and test sets have the same label sets. Thus, samples from external classes of a client are OOD samples of that client. 2) In heterogeneous FL, different clients have different label sets. For example, one client can have a label set of {0,1}, which means it is trained on classes 0 and 1. In this case, this client’s test set only contains samples from classes 0 and 1, since the utility of this client is to classify between class 0 and 1. Samples from other classes (e.g., classes 2-9) are OOD samples for this client. In other words, different clients can have different classes as in-distribution and OoD classes.
>
> **2. FOSTER is based on personalized FL. What about the baselines? The paper only mentions that the training method of all approaches is FedAvg in Section 5. The baselines should also adopt personalized FL for fair comparison.**
>
> Thank you very much for your comment. We agree that FOSTER considers personalized FL, for fair comparison, both the baselines and FOSTER adopt FedAvg with personalized classifier head. In the revised manuscript, we clarified this point in baseline settings of section 5.
>
> Besides, we would also like to clarify that the proposed FOSTER is a generic algorithmic framework that can be incorporated into different FL algorithms and that FedAvg is just one of FL instances we use for illustration. We have added experiments to show FOSTER performance when combined with a representative personal FL approach FedProx [A]. Our results in Table 11 (Appendix) show that FOSTER outperforms competitive OoD baselines in both FL instances.
>
> [A] Li T, Sahu A K, Zaheer M, et al. Federated optimization in heterogeneous networks[J]. Proceedings of Machine Learning and Systems, 2020, 2: 429-450.
>
> **3.  Local training without FL can also be added as a baseline, e.g., the clients can simply conduct unsupervised learning locally such as one-class SVM.**
>
> We would also thank the reviewer for pointing out the baseline of local training without FL. In this revised version we added one-class SVM for local training without FL as a baseline, we report the results in Table 12 in the appendix. We see that the proposed approach can significantly outperform the one-class SVM in terms of  AUROC and AUPR  for OoD detection.
>
> **4. Figure 2 is confusing. The paper should clearly demonstrate why external-class data can help in distinguishing the ID and OoD data.**
>
> Thank you very much for your comment. We have revised the figure and manuscript (in blue) to address this confusion. We replace Fig.2 using the density of negative energy score, to better explain the intuition that external-class data are natural OoD data that can be used for OoD training.
>
> Specifically, we evaluate the energy score after **normal training**, **training w/ VOS**, and **training w/ external-class data**. Fig.2 shows that incorporating external-class data can effectively increase the margin between real OoD and ID data (Fig.2 (c)), since external-class data can serve as real OoD surrogate samples.  Besides, Fig.2 (a)-(b) also verifies that external-class samples are likely to be nearer to the ID data.
>
> To further demonstrate why external-class data can help OoD training, we also conduct central experiments without FL training, which is shown in A.5 central experiments: ‘Effects of the number of ID classes’. We fix the training data size to be 15000, and vary ID classes number to be 7, 5, and 3. According to the results shown in Table 13, with limited ID samples from each class, external-class data outperforms other baselines for OoD detection without hurting ID Acc.
>
> **5. Experiments to investigate the effect of client sampling and data heterogeneity can be added.**
>
> To investigate the effect of different client sampling, in section 5.3: ‘Effects of active client number’, we have evaluated the performance of our proposed method and competitive baselines w.r.t. different numbers of sampled active clients for each round, and FOSTER is not sensitive to different client sampling.  To investigate the effect of data heterogeneity, in section 5.3: ‘Effects of ID class number’, we have evaluated the performance of our proposed method and competitive baselines w.r.t. different ID class numbers for each client. With different ID class numbers, the dissimilarity between different clients will be different, which varies the data heterogeneity among clients. Different data heterogeneity has a large impact on VOS, however, it has no effect on FOSTER.

---

> ### Author Response · Authors · 2022-11-18
> **Thank you for your helpful comment and suggestions - Part 2**
>
> We address the rest of your concerns as follows:
>
> **6. A generator is required in FOSTER, which introduces additional privacy concern, communication, and computation overhead. The paper should discuss these aspects.**
>
> Thank you very much for your comment. We have added related content to discuss communication and computation overhead, and privacy concern. Specifically, we discuss communication costs, including per-round communication costs and the number of communication rounds, in the appendix section A.3 ‘Communication cost for FOSTER’.
>
> For privacy the concern, we would like to emphasize that many existing generative/synthesizer-based methods [A-C] for heterogeneous FL also have the privacy overhead and FOSTER does not introduce additional privacy overhead as compared to these methods. Our work pioneers the first study on the OoD detection in class non-iid FL, following the common privacy practice of prior arts [A-C]. There have been many studies regarding privacy-protection generators [D-E], which can be integrated with our method to enhance the privacy protection by injecting noise into the gradients for the generator.
>
> [A] Zhang L, Shen L, Ding L, et al. Fine-tuning global model via data-free knowledge distillation for non-iid federated learning. CVPR 2022.
>
> [B] Zhu Z, Hong J, Zhou J. Data-free knowledge distillation for heterogeneous federated learning[C]//ICML2021.
>
> [C] Li W, Chen J, Wang Z, et al. IFL-GAN: Improved Federated Learning Generative Adversarial Network With Maximum Mean Discrepancy Model Aggregation[J]. TNNLS, 2022
>
> [D] Jordon J, Yoon J, Van Der Schaar M. PATE-GAN: Generating synthetic data with differential privacy guarantees[C]//ICLR. 2018.
>
> [E] Xu C, Ren J, Zhang D, et al. GANobfuscator: Mitigating information leakage under GAN via differential privacy[J]. IEEE Transactions on Information Forensics and Security, 2019, 14(9): 2358-2371.
>
>
> **7. Quality: The paper is in a good shape but not solid enough as it lacks in-depth analyses of the effectiveness of the proposed approach.**
>
> We have included new empirical results to strengthen our analyses of the effectiveness of our approach, and we believe these new analyses can greatly improve our understanding of its effectiveness. Specifically, we have conducted extensive analysis on the intuition that external-class samples can help distinguishing OoD and ID samples in section 4.1, and also added central experiments without FL training, in section A.5 in the appendix to provide more support and explanation of our intuition and conclusions. Further, we also add a communication cost analysis of the proposed approach in section A.3 in the appendix.
>
> **8. Novelty: The idea of using GAN to help the training is not new [1].**
>
> [1] FedCG: Leverage Conditional GAN for Protecting Privacy and Maintaining Competitive Performance in Federated Learning.
>
> We appreciate the pointer and added it to our paper as reference. Though the use of generator in FL or even central learning is not new, our method essentially differs from [1] on purpose and technique, forming our non-neglectable novelty.
>
> We are the first to use the generator for modeling out-of-distribution data in federated learning. In contrast, traditional methods including [1] only consider to use the generator to model in-distribution data.
>
> Because of the distinct purpose, we introduce several novel techniques. First, we use the generator samples for OoD loss instead of cross-entropy loss [1]. Second, we introduce a novel filtering strategy to encourage diversity and enlarge the margin between the ID and synthetic OoD data. In comparison, [1] generate samples that are overlap with the local distribution and directly apply all the generated samples for training the in-distribution classifier.
>
> Other than the generator,  our paper has a multifold of novelty. Our work is the first solution of tackling OoD detection in federated learning (FL). The detection of OoD samples nowadays have been essential to protect the integrity of the predictive model , and yet the data confidentiality requirement of FL and limited sample size of FL clients have imposed critical challenges of OoD detection in FL. This challenge prevents directly plugging in previous OoD detection techniques, as we have illustrated in our paper and empirical studies. We provide a **novel principle** that turns the curse of data heterogeneity into a blessing that facilitates OoD detection.Our work bridges a critical research gap since OoD detection for FL is currently not yet well-studied in the literature. To our knowledge, the proposed FOSTER is the first OoD learning method for FL that does not require real OoD samples.
>
> **9. Reproducibility: Some experimental details are missing**
>
> We have revised the paper to provide more details for reproducibility. Some of the experimental settings are provided in appendix A.1.

---

> > ### Comment · Reviewer_3v8K · 2022-11-21
> > **Thanks for your response**
> >
> > Your responses 1-6 are clear and address most of my concerns. I have increased my score to 6.
> >
> > I realize that I had misunderstandings about the OoD setting. Now I have a new concern about the experimental setting. The paper simulates an ideal federated setting in the experiments, where each client has a fixed number of classes. If the paper is accepted, I hope you can consider more practical federated scenarios/partition strategies (e.g., partitioning by Dirichlet distribution [1]) to demonstrate the effectiveness of the proposed approach in the final version.
> >
> > [1] Measuring the Effects of Non-Identical Data Distribution for Federated Visual Classification

---

> > > ### Author Response · Authors · 2022-11-23
> > > **Thank you for reading our response**
> > >
> > > Thank you very much for carefully reading our response and increasing your score! We are glad our response has addressed most of your concerns, and we address your concern about experimental setting as follows:
> > >
> > > **Now I have a new concern about the experimental setting. The paper simulates an ideal federated setting in the experiments, where each client has a fixed number of classes. If the paper is accepted, I hope you can consider more practical federated scenarios/partition strategies (e.g., partitioning by Dirichlet distribution [1]) to demonstrate the effectiveness of the proposed approach in the final version.**
> > >
> > > [1] Measuring the Effects of Non-Identical Data Distribution for Federated Visual Classification
> > >
> > > Thank you for the detailed comments! We conduct one more experiments on STL10 with Dirchlet partition. The results are shown in the following table, and we will update the full results into the final revision. According to the results, FOSTER still outperforms baselines given Dirichlet partition.
> > >
> > >  |Method|AUROC|AUPR|ACC|
> > > |  ----  | ----  |  ----  | ----  |
> > >  |MSP|0.6894|0.9269|0.6608|
> > >  |Energy|0.7481|0.9310|0.6608|
> > >  |Odin|0.6901|0.9270|0.6608|
> > >  |VOS|0.7455|0.9128|0.6628|
> > > | FOSTER|**0.7724**|**0.9387**|**0.6980**|

---

### Official Review · Reviewer_Liwm · 2022-10-24

**Confidence:** 3
**Correctness:** 2
**Technical Novelty And Significance:** 2
**Empirical Novelty And Significance:** 2
**Recommendation:** 6

**Clarity, Quality, Novelty And Reproducibility:**

This paper is well-structured and clearly written. However, I do not think this paper is self-contained, for examples, descriptions about the MSP and VOS are missing. The novelty of this paper, at least to me, is limited. Further, I did not check the reproducibility of the paper.

**Strength And Weaknesses:**

> Strength

- The authors consider the problem of OOD detection in federated learning, which seems to be a realistic problem. For security sensitive applications such as autonomous driving and voice recognition authorization, I agree with the authors that FL can be useful in these scenarios and taking OOD detection into consideration can be important.

- The authors conduct a set of experiments with various OOD detection settings, e.g., CIFAR benchmarks, STL10, and hard OOD detection. To some extent, the experimental results verify the superiority of the proposed method over a set of classical OOD detection methods.

> Weakness

- *The literature review is not enough*. The authors attribute the existing methods in OOD detection into two classes, namely, the real-data approaches and the synthetic approaches. However, I believe there is a large group of methods that do not rely on any OOD data (either real ones or the virtual ones), which is uncovered in the authors’ discussion. These works (e.g., post-hoc approaches [1], fine-tuning approaches [2], and contrastive learning approaches [3]) are popular and effective in OOD detection (even better than OE in many cases). Since they do not rely on OOD data for training, it seems that the considered problem of OOD data scarcity is actually not a big issue. I think the authors should cover a larger group of methods, and then discuss why collecting/synthesizing additional OOD data is important in FL.

[1] Yiyou Sun, et al. Out-of-distribution Detection with Deep Nearest Neighbors. ICML’22.

[2] Haoqi Wang, et al. ViM: Out-of-distribution with Virtual-logit Matching. CVPR’22.

[3] Vikash Sehwag, et al. SSD: A Unified Framework for Self-supervised Outlier Detection. ICLR’21.

- The authors follow the methodology named VOS. When applying the VOS for FL, the authors claim that enormous examples should be drawn for an accurate estimation of parameters. However, *I am not sure if the statement is true*. Especially, the estimation of MoG is in low dimensional space and the number of tunable parameters are in small scale. In fact, the authors of VOS actually claimed their superiority over previous GAN-based methods [4] in easy to be optimized, which may refute this paper’s point. Therefore, I think further discussion and experimental justification may require here.

[4] Kiin Lee, et al. Training Confidence-calibrated Classifiers for Detecting Out-of-distribution Samples. ICLR’18.

- Another little confusion point is that *if the data are synthesized in low-dimensional space where the MoG can properly model it, is it necessary to use the generative models in data synthesis?* It may increase the number of trainable parameters, which may exaggerate the data scarcity issue in FL.







**Summary Of The Paper:**

The authors studied the problem of OOD detection in the literature of federated learning. The authors claim that the main challenge that prevents previous state-of-the-art OOD detection methods from being incorporated to FL is that they require large amount of real OOD samples, which are closely or even infeasible to obtain in reality. Further, the authors claimed that the data heterogeneity where each client collects non-iid data can hurt the performance of the system. Accordingly, the authors suggest to so called “taking advantage of such heterogeneity and turning the curse into a blessing that facilitates OOD detection”. Specially, the authors propose a novel federated OOOD Synthesizer (FOSTER), which learns a class-condtional generator to synthesize virtual external-class OOD samples. The authors claimed their superiority over the state-of-the-art counterparts.

**Summary Of The Review:**

I think studying OOD detection in the literature of FL is an interesting problem, but I am not sure if OOD data scarcity is really an important problem that require the in-depth discussion.

---

> ### Author Response · Authors · 2022-11-18
> **Thanks for your helpful comments and suggestions-Part 1**
>
> Thank you for your comment and we address your concerns as follows:
>
> **1. The literature review is not enough. The authors attribute the existing methods in OOD detection into two classes, namely, the real-data approaches and the synthetic approaches. However, I believe there is a large group of methods that do not rely on any OOD data (either real ones or the virtual ones), which is uncovered in the authors’ discussion. These works [1-3] are popular and effective in OOD detection. Since they do not rely on OOD data for training, it seems that the considered problem of OOD data scarcity is actually not a big issue. I think the authors should cover a larger group of methods, and then discuss why collecting/synthesizing additional OOD data is important in FL.**
>
> [1] Yiyou Sun, et al. Out-of-distribution Detection with Deep Nearest Neighbors. ICML’22.
>
> [2] Haoqi Wang, et al. ViM: Out-of-distribution with Virtual-logit Matching. CVPR’22.
>
> [3] Vikash Sehwag, et al. SSD: A Unified Framework for Self-supervised Outlier Detection. ICLR’21.
>
> We categorized the existing methods into two classes, post hoc, and methods during training (see ``related work``). All post hoc methods do not rely on real OoD data. Thanks for the references, and we have incorporated them in our literature review with discussions and added experimental results. Specifically, we add [1] and [2] to the post hoc category and added [3] to the training category that does not require real OoD data.
>
> We have added [1][2] to our competitive baselines and the results show in Table 12 in the appendix. According to the results reported in Table 12, both [1] and [2] cannot achieve comparable results to VOS, FOSTER and other post hoc scores in heterogeneous FL.
>
> **2. The authors follow the methodology named VOS. When applying the VOS for FL, the authors claim that enormous examples should be drawn for an accurate estimation of parameters. However, I am not sure if the statement is true.**
>
> We would like to point out that VOS performs worse in FL setting with limited local training data. To give a demonstration, we conduct central experiments without FL training on CIFAR10 with limited samples from each class and training data size equal to 15000. The experiment setting and results are shown in appendix A.5 central experiments. According to the results in Table 13, with limited samples for each class in the training set, VOS shows lower AUROC and AUPR compared with the results reported in [A] where the entire training set is utilized, and worse than using external-class data. Thus, VOS does need enormous examples for accurate estimation.
>
> [A] Du X, Wang Z, Cai M, et al. VOS: Learning What You Don't Know by Virtual Outlier Synthesis[J]. arXiv preprint arXiv:2202.01197, 2022.
>
> **3. The authors of VOS actually claimed their superiority over previous GAN-based methods [4] in easy to be optimized, which may refute this paper’s point. Therefore, I think further discussion and experimental justification may require here.**
>
> [4] Kiin Lee, et al. Training Confidence-calibrated Classifiers for Detecting Out-of-distribution Samples. ICLR’18.
>
> Thank you very much for your comment. First, the purpose of GAN [4] is completely different from the generator for FOSTER, since they are based on two different assumptions.  [4] generates ‘boundary’ samples in the low-density area of ID data in central mode, while we use generator with a global classifier head to generate external-class OoD data in heterogeneous FL.
>
> Second, our proposed method leverages both a generator to synthesize external class data and VOS-like estimation to collaboratively achieve the goal of OoD, instead of claiming that all generator-based methods outperformed VOS. Thus, this paper will not refute the conclusion in VOS.
>
> **4. If the data are synthesized in low-dimensional space where the MoG can properly model it, is it necessary to use the generative models in data synthesis? It may increase the number of trainable parameters, which may exaggerate the data scarcity issue in FL.**
>
> Thank you very much for your comment. We agree that MoG uses fewer trainable parameters, however, MoG requires real samples for estimation, which is prohibited due to the data confidentiality requirement of FL. Without access to real external-class data, we can adopt a generator and a global classifier first to synthesize virtual external-class samples before the modeling of MoG.

---

> ### Author Response · Authors · 2022-11-18
> **Thank you for your helpful comments and suggestions - Part 2**
>
> Thank you for your positive comment about our paper, and we address your concern of clarity, novelty, and reproducibility as follows:
>
> **1.  I do not think this paper is self-contained, for examples, descriptions about the MSP and VOS are missing.**
>
> In the revised version,  the detailed description of MSP and VOS are included in section 2 ‘related work’ to make the paper self-contained.
>
> **2. Novelty**
>
> We would like to emphasize our major contribution, which is the first solution of tackling OoD detection in federated learning (FL). The detection of OoD samples nowadays have been essential to protect the integrity of the predictive model (especially the deep ones), and yet the data confidentiality requirement of FL and limited sample size of FL clients have imposed critical challenges of OoD detection in FL. This challenge prevents directly plugging in previous OoD detection techniques, as we have illustrated in our paper and empirical studies. Besides, our novel and elegant solution takes advantage of heterogeneity in FL and turns the curse into a blessing that facilitates OoD detection. By using external-class data as OoD samples, our proposed approach breaks all the curses and do not need any additional OoD data.
>
> **3. Reproducibility**
>
> We have revised the paper to provide more details for reproducibility. Some of the experimental settings are provided in appendix A.1. We promise to release codes upon acceptance.

---

> > ### Comment · Reviewer_Liwm · 2022-11-22
> > **Thank you for the response**
> >
> > The authors conduct more experiments about post-hoc approaches in the considered setup, demonstrating the superiority of their proposal over the state-of-the-art baselines. The authors also largely address my concerns about the algorithm design, so I would like to raise my score to 6.

---

> > > ### Author Response · Authors · 2022-11-23
> > > **Thank you for reading our response**
> > >
> > > Thank you very much for carefully reading our response and increasing your score! We are glad our response has addressed your concerns.

---

### Official Review · Reviewer_eA7B · 2022-10-25

**Confidence:** 5
**Correctness:** 4
**Technical Novelty And Significance:** 4
**Empirical Novelty And Significance:** 4
**Recommendation:** 8

**Clarity, Quality, Novelty And Reproducibility:**

This paper is clearly demonstrated. The quality and novelty are enough in terms of significance and technical contributions. This paper can be reproducible based on the algorithm provided.

**Strength And Weaknesses:**

Pros:

1.  The problem setting is very important, filling up a gap between FL and OOD detection. This study is significant in the fields of FL and OOD detection.

2. From the technical part, the authors propose a novel federated OOD synthesizer to take advantage of data heterogeneity to facilitate OOD detection in FL, allowing a client to learn external class knowledge from other non-iid federated collaborators in a privacy-aware manner. This work bridges a critical research gap since OOD detection for FL is currently not yet well-studied in literature. The proposed FOSTER is the first OOD learning method for FL that does not require real OOD samples. Note that, it is not trivial to directly use the OOD techniques in FL, which is the major technical contribution of this paper.

3. Experiments cover many aspects regarding this paper (like CIFAR10/100 and near OOD detection).


Cons:

1. The problem setting should be presented explicitly. What the data you have and what the aim this paper wants to do should be demonstrated in a separated paragraph or subsection.

2. The generated OOD data is somehow different from the true OOD data. Relevant discussions are needed.

3. What is the validation dataset used to find the best hyperparameters of your method? Validation datasets are very important to the OOD detection. We cannot select validation datasets containing OOD data.

4. ImageNet benchmark should be used to verify the effectiveness of the proposed method.

5. To ensure the completeness of the ablation study, performance of Foster w/o pdf and soft label should be reported.

6. In section 3, it is wired to say that a sample belongs to (using \in) a distribution. For example, x \in D. It should be x ~ D.




**Summary Of The Paper:**

In this paper, the authors study a largely overlooked problem: OOD detection in FL. To turn the curse of heterogeneity in FL into a blessing that facilitates OOD detection, the authors propose a novel OOD synthesizer without relying on any real external samples, allowing a client class knowledge from other non-iid federated collaborators in a privacy-preserving manner. Empirical results showed that the proposed approach achieves SOTA performance in non-iid FL.

From the view of the problem setting, this paper studies a very important problem, which contains enough significance in the field of FL and OOD detection. From the technical part, the authors propose a novel federated OOD synthesizer to take advantage of data heterogeneity to facilitate OOD detection in FL, allowing a client to learn external class knowledge from other non-iid federated collaborators in a privacy-aware manner. This work bridges a critical research gap since OOD detection for FL is currently not yet well-studied in literature. The proposed FOSTER is the first OOD learning method for FL that does not require real OOD samples.


**Summary Of The Review:**

From the view of the problem setting, this paper studies a very important problem, which contains enough significance in the field of FL and OOD detection. From the technical part, the authors propose a novel federated OOD synthesizer to take advantage of data heterogeneity to facilitate OOD detection in FL, allowing a client to learn external class knowledge from other non-iid federated collaborators in a privacy-aware manner. This work bridges a critical research gap since OOD detection for FL is currently not yet well-studied in literature. The proposed FOSTER is the first OOD learning method for FL that does not require real OOD samples.

---

> ### Author Response · Authors · 2022-11-18
> **Thanks for your helpful comments and suggestions**
>
> We are glad that the reviewer found our problem setting important and our method novel. We thank the reviewer for the constructive comments and suggestions, which we address below:
>
> **1.The problem setting should be presented explicitly.**
>
> Thank you very much for your comment. We have made the problem setting explict in this revision. At the beginning of section 3 of the revised manuscript, we give the problem setting in a separate paragraph as follows (marked in blue):
>
> ``In this paper, we consider classification tasks in heterogeneous FL settings, where non-iid clients have their own label set for training and testing samples. Our goal is to achieve OOD-awareness on each client in this setting.``
>
> Following the paragraph, we formally state the problem setting regarding the data and learning objectives.
>
> **2. Relevant discussions for the generated OoD data**
>
> Thank you raising this and we have clarified this issue in the paper at section 5.1.
>
> The purpose of OoD data is to provide a reference of samples outside of the distribution of interest and therefore there are many choices of OoD samples but good ones should have distinct distribution from that of in-distribution samples. According to Figure 3. The generated virtual OoD samples are strictly out of the local ID distribution without any overlap. This shows that they are good substitutes for real OoD samples during training to benefit OoD detection.
>
> **3. What is the validation dataset used to find the best hyperparameters of your method? Validation datasets are very important to the OOD detection. We cannot select validation datasets containing OOD data.**
>
> Thank you very much for your comment. We do not have an OoD dataset for validation, but we conduct an parameter sensitivity study evaluating different values of hyperparameters. We use a hyperparameter that shows good performance in the parameter study, discussed in Appendix A.2.
>
> **4. About ablation study, performance of Foster w/o pdf and soft label should be reported.**
>
> Thank you very much for your comment. We followed your nice suggestions and added corresponding experimens. The ablation study of w/ and w/o pdf filter has been shown in Table 6, and the ablation study of w/ and w/o soft label is in Table 7. To understand how pdf filter increases sample diversity, we add Table 10 in the appendix to show the variance of the ID p.d.f of selected samples of filter for three different clients.
>
> **5. In section 3, it is wired to say that a sample belongs to (using \in) a distribution. For example, x \in D. It should be x ~ D.**
>
> Thank you for pointing out and we have revised this problem in the updated manuscript.
>
> **6. ImageNet benchmark.**
>
> Since it takes us a long time to conduct FL simulation experiments on ImageNet benchmark with 1000 classes, and we only have limited revision time, we conduct our experiments on ImageNet benchmark with 12 classes. We Use 50 clients, and 3 classes for each client. The results for ImageNet Benchmark is shown as follows:
>
> |Method|Acc|AUROC|AUPR|
> |  ----  | ----  |  ----  | ----  |
> |Energy|0.8552|0.7267|0.9174|
> |MSP|0.8552|0.7447|0.9286|
> |ODIN|0.8552|0.7399|0.9277|
> |VOS|0.8605|0.7207|0.9171|
> |FOSTER|**0.8663**|**0.7526**|**0.9312**|

---

> > ### Comment · Reviewer_eA7B · 2022-11-22
> > **My concerns are well addressed**
> >
> > I will keep my original score since my concerns are addressed well. I recommend accepting this paper.

---

> > > ### Author Response · Authors · 2022-11-23
> > > **Thank you for reading our response**
> > >
> > > Thank you very much for carefully reading our response! We are glad our response has addressed your concerns.

---

### Decision · Program_Chairs · 2023-01-20

**Decision:**

Accept: notable-top-25%

**Justification For Why Not Higher Score:**

I am not sure how many oral presentations we will have, but perhaps this submission is not that competitive to make an oral presentation.

**Justification For Why Not Lower Score:**

The negative side of a task can actually be the positive side of another task, if we switch the problem of interest and change our point of view accordingly. This is quite interesting and may inspire a lot of researchers/practitioners attending ICLR 2023, and hence the paper should be seen by more people as a spotlight presentation.

**Metareview: Summary, Strengths And Weaknesses:**

The paper studied out-of-distribution detection problem under the federated learning setting. After switching the problem of interest from classification of in-distribution data to detection of out-of-distribution data, it is interesting (if not surprising) to see that heterogeneity of data becomes a positive point rather than a negative point of training data. The philosophy is to view in-distribution data from other clients as out-of-distribution data for the current client, and the implementation is to train a class-conditional generator (called Federated Out-of-Distribution Synthesizer) in a federated manner. All reviewers agreed that the novelty and significance are above the bar and thus we should accept it for publication.

**Note From Pc:**

if the above contains the word "oral" or "spotlight" please see: "oral" presentation means -> notable-top-5% and "spotlight" means -> notable-top-25%. As stated in our emails, we are disassociating presentation type from AC recommendations